# Nanoscatterer-Assisted Fluorescence Amplification Technique

**DOI:** 10.3390/nano13212875

**Published:** 2023-10-30

**Authors:** Sylvain Bonnefond, Antoine Reynaud, Julie Cazareth, Sophie Abélanet, Massimo Vassalli, Frédéric Brau, Gian Luca Lippi

**Affiliations:** 1Université Côte d’Azur, UMR 7010 CNRS, Institut de Physique de Nice, 06560 Valbonne, France; sylvain.bonnefond@oberon.one; 2Université Côte d’Azur, UMR 7275 CNRS, Institut de Pharmacologie Moléculaire et Cellulaire, 06560 Valbonne, France; reynaud@ipmc.cnrs.fr (A.R.); cazareth@ipmc.cnrs.fr (J.C.); abelanet@ipmc.cnrs.fr (S.A.); brau@ipmc.cnrs.fr (F.B.); 3James Watt School of Engineering, University of Glasgow, Glasgow G12 8LT, UK; massimo.vassalli@glasgow.ac.uk

**Keywords:** nanoparticles, multiple scattering, light amplification, optical gain, linewidth narrowing, stimulated emission, fluorescence

## Abstract

Weak fluorescence signals, which are important in research and applications, are often masked by the background. Different amplification techniques are actively investigated. Here, a broadband, geometry-independent and flexible feedback scheme based on the random scattering of dielectric nanoparticles allows the amplification of a fluorescence signal by partial trapping of the radiation within the sample volume. Amplification of up to a factor of 40 is experimentally demonstrated in ultrapure water with dispersed TiO${}_{2}$ nanoparticles (30 to 50 nm in diameter) and fluorescein dye at 200 μmol concentration (pumped with 5 ns long, 3 mJ laser pulses at 490 nm). The measurements show a measurable reduction in linewidth at the emission peak, indicating that feedback-induced stimulated emission contributes to the large gain observed.

## 1. Introduction

Fluorescence is the radiative component of the spontaneous relaxation of an emitter (typically a molecule) from an excited state characterized by a spectral distribution that identifies its nature. Stimulated by the absorption of light at a shorter wavelength, it has been extensively studied and applied for over a century and has found countless applications. Its practical use has increased dramatically with the development of sophisticated fluorescence-based techniques and the availability of a wide range of fluorescent probes and markers, benefiting various fields of investigation and monitoring [1]. In chemistry, fluorescence spectroscopy has become a standard analytical tool for the study of molecular structures, interactions and chemical kinetics [2,3]. Fluorescent probes have been developed to bind selectively to specific targets, allowing the sensitive detection and imaging [4] of biological molecules in cellular and tissue samples [5,6,7,8]. Nanoparticle plasmonic resonances are being studied as amplification mechanisms for fluorescence [9].

Fluorescence has also made significant contributions to materials science and nanotechnology. Quantum dots, a class of semiconductor nanoparticles with tunable fluorescence properties, have enabled breakthroughs in quantum information processing and biomedical imaging [10]. Fluorescence-based sensors and nanomaterials have been developed for applications ranging from environmental monitoring [11] to medical diagnostics [12,13]. Fluorescent probes have been used to detect and quantify pollutants, monitor water quality and assess the health of ecosystems [14,15,16]. Food monitoring has received much attention due to health and safety issues [17] and resource conservation [18]. Fluorescence methods have successfully contributed to its development [19,20,21].

Based on the same theoretical foundations of lasers, we consider a new solution to improve the efficiency of fluorescence emission in a less constrained environment. It is based on the physical principle of random lasers and the multiple scattering of light [22,23]. Early work [24] laid the foundations for stimulated amplification by incoherent positive feedback from scatterers in a diffuse regime, known at the time as a “photonic bomb”. Its first experimental demonstration replaced one mirror of a Fabry–Perot cavity with a diffusive surface [25]. Many different random lasers have now been described [23] using solutions of Rhodamine 6G (Rh6G) as a gain medium, which is a cytotoxic dye [26] usually diluted in non-biocompatible organic solvents or at a non-physiological pH [27]. Using the intrinsic architecture of a biological tissue as a natural scatterer, one team described the use of Rh6G to generate random lasers from bone fibers stained with this dye [28]. Light localization in nanostructures [29] and biomaterials [30] have also been studied for achieving light amplification, and new applications of random lasers are focusing on sensing [31].

Combining the concepts of biological [32] and random lasers [23], the aim of this work is to lay the foundations for the sub-laser threshold fluorescence amplification of biological samples, while keeping the experimental conditions as close as possible to those of a biological environment, thus enabling a broad field of potential applications. This scope sets our contribution apart from other successful investigations (see [33,34]), where lasing has been achieved by using fluorophores whose chemical properties may be incompatible with numerous applications (e.g., living systems). In our scheme, a stimulated emission fraction is generated by recycling the excitation and emission photons thanks to scattering in the sample (weak localization).

In this study, we demonstrate the possibility of obtaining significant stimulated fluorescence enhancement from a fluorophore commonly used in cell biology, in an aqueous medium and at a biological pH. Working with biological samples requires careful consideration of suitable fluorophores, potential photodissociation, photobleaching or phototoxicity induced by optical pumping and by scatterers added to the sample. Some of these constraints apply also to other fields but are typically not all simultaneously present; their concurrent fulfillment ensures a broader potential for applying the technique. We note that the amplification obtained in the course of this work also leads to a spectral narrowing of the fluorescence, thus adding to the detection of intrinsically weak fluorescence signals the advantage of denser multiplexing of fluorochromes for (e.g., biomarker) parallel identification [35]. After discussing the choices made (Section 2), we describe the sample preparation (Section 2.3.1), the experimental setup (Section 3) and its calibration (Section 3.5), followed by the techniques used for data processing (Section 4) and an analysis of the results (Section 5).

## 2. Materials and Methods

The aim of this experimental work is to present a flexible amplification technique that can be applied in different fields. Therefore, instead of choosing the conditions that may give the best results but are not necessarily widely applicable, we choose average conditions that allow a better assessment of the potential usefulness of our proposal. The medium in which we test amplification is water, so to obtain good amplification, the refractive index of the NanoParticles (NPs) must be compared with that of this medium. Changing the medium will require rescaling.

### 2.1. Materials

#### 2.1.1. Fluorophore

We choose Fluorescein-5-Isothiocyanate (FITC), a broadly used fluorophore in biological applications—traditional and innovative [36]—with good overall performance, ease of finding and manipulating (no health risks) and environmental friendliness. In spite of its overall good performance (quantum yield and brilliance), it is limited in its cycling properties (bleaching takes place in <10${}^{6}$ cycles). The choice of a mid-range fluorophore with good average performance reflects the overall philosophy of the study.

#### 2.1.2. Scatterers

Titanium dioxide nanoparticles (TiO${}_{2}$-NPs) have the advantage of being readily available, at a very reasonable cost, and with a low environmental impact (apart from the usual precautions required to manipulate NPs). Indeed, they are widely used in numerous contexts, and although their biocompatibility has been questioned [37], submicron-sized particles (including nano-sized fractions) of TiO${}_{2}$ have been used in food and cosmetics as a pigment for human use for more than 50 years. In addition, TiO${}_{2}$ NPs absorb only in the UV region of the spectrum and are therefore compatible with the optimal pump wavelength for FITC (λ = 490 nm). For these reasons, we select TiO${}_{2}$-NPs as an excellent candidate for testing the amplification technique.

The rutile form of titanium dioxide nanoparticles (TiO${}_{2}$-NPs) was chosen because of its higher refractive index ($n_{rutile}=2.87$ @ 500 nm [38]) compared to the anatase form ($n_{anatase}=2.56$ @ 500 nm [39]). The high index contrast, relative to the surrounding environment (mostly water with n≈1.33 [40,41], i.e., relative refractive index $\frac{n_{{\mathrm{TiO}}_{2}}}{n_{\mathrm{water}}}\approx 2.16$), ensures greater light-scattering strength [27] within the gain medium. The TiO${}_{2}$-NPs play the role of passive elastic scatterers and lengthen the effective optical path of the radiation (both pump and fluorescence), thereby promoting the amplification of the fluorescence process [27,42,43].

### 2.2. Optical Pumping

As one of the mechanisms to obtain amplification relies on achieving the stimulated emission regime, FITC pumping is performed with a pulsed laser, as is common in the literature [23]. The high photon flux is indeed necessary to achieve a sufficient photon density in the excited volume to achieve amplification by stimulated emission. However, to reduce potential damage from the pump (photobleaching and potential phototoxicity), we exploit the short and powerful pulses delivered by a Q-switched laser, separated by long waiting times (low repetition rates) typical of many solid-state devices. This keeps the total exposure of the sample to a low level.

Considering pump pulses with energy in the range of $E_{p}\approx 5$ mJ and duration $\tau_{p}\approx 5$ ns, we obtain peak pump power values $P_{p}\approx 10^{6}$ W, i.e., a peak photon flux $\Phi_{p}\approx 2.5\times 10^{24}$ s${}^{-1}$ and an integrated dose per pulse $N_{ph}\approx 1.3\times 10^{16}$ photons. For a pulse repetition rate of $\nu_{p}=10$ Hz (Section 3), the exposure duty cycle is $\delta =\tau_{p}\cdot \nu_{p}\approx 5\times 10^{-8}$, resulting in an average number of photons $\langle N_{ph}\rangle =N_{ph}\cdot \delta \approx 6.5\times 10^{8}$. Compared to exposure with a continuous wave (cw) laser, this would be equivalent to fluorescence experiments with $P_{cw}\approx 2.5$ nW, well below the standard fluence where the laser power is typically in the mW range.

### 2.3. Methods

#### 2.3.1. Sample Preparation

To a solution of FITC (F6377, Sigma-Aldrich, Burlington, MA, USA, [44]) in ultrapure water (H${}_{2}$O mQ) @ pH7, concentration $C_{F}=200$ μM, we add rutile TiO${}_{2}$-NPs (7013WJWR, NanoAmor, Nanostructured & Amorphous Materials, Inc., Katy, TX, USA [45]), at concentrations $C_{N}=\left(1.56,3.12,6.25\right)$ mg/mL.

#### 2.3.2. Fluorescein

Having chosen a good but not optimal fluorophore, we analyse the performance of its dilutions to obtain low self-quenching [1] at moderate concentrations. Fluorescence emission spectra of increasing $C_{F}$ in H${}_{2}$O mQ were recorded (spectrofluorimeter FP-8300 JASCO, Easton, MD, USA). Figure 1a shows the maximum emission intensity at wavelength $\lambda_{F}\approx 520$ nm as a function of $C_{F}$, showing that the strongest fluorescence intensity is obtained at $C_{F}=200$ μM. When $C_{F}>200$ μM (Figure 1a), the fluorescence intensity is reduced, probably due to the self-quenching resulting from photon re-absorption by the dye [1].

#### 2.3.3. TiO${}_{2}$

$C_{N}$ was chosen to match the diffusive regime of light scattering: $L>>l_{N}>>\lambda$ [42] where *L* is the sample thickness (2 mm, Section 3.3), λ is the wavelength (≈500 nm) and $l_{N}$ is the scattering mean free path. The latter can be expressed as a function of the particle mass concentration $C_{N}$ and the scattering cross section $\sigma_{N}$, $l_{N}=1/\left(C_{N}\cdot \sigma_{N}\right)$ [27,42], and takes the numerical values $l_{N}=\left(245,123,61\right)\;\mu$m for the concentration values on which we focus in the experiment ($C_{N}=\left(1.56,3.12,6.25\right)$ mg/mL).

### 2.4. Scatterer Characterization

The mean diameter of TiO${}_{2}$-NPs is (30<〈D〉<50) nm (manufacturer’s specification [45]) when supplied in their liquid suspension (H${}_{2}$O, CAS#7732-18-5). We choose this range of TiO${}_{2}$-NPs as a compromise to keep the scattering as isotropic (and polarization-independent) as possible, while maintaining a sufficiently large scattering coefficient. However, electrostatic forces generally intervene when the sample is transferred to an ionic solution—a common occurrence in numerous applications—and since the scattering characteristics depend sensitively on the size of the scatterers, reproducibility requires obtaining a stable suspension. In fact, charge-induced clustering has several shortcomings for efficient amplification: (i) larger effective particles, resulting in an overall reduction in the scattering amplitude [42]; (ii) lower density of the resulting scatterers, reducing the number of secondary radiation sources; and (iii) larger mass, hence the rapid precipitation of the suspension [46]. The latter is particularly important for TiO${}_{2}$-NP due to their high density ($\mu_{N}=4.23\times 10^{3}$ Kg/m${}^{3}$). Care must therefore be taken to obtain a stable, cluster-free solution.

#### 2.4.1. ζ-Potential

The physico-chemical equilibrium of NPs is ensured by the mutual repulsion [47,48], quantified by the isoelectric potential [49,50], which of course depends on the pH of the solution. The sample’s isoelectric point results from the manufacturing process and therefore varies from one manufacturer to another, with consequent differences in the surface and chemical behavior [46,49,50]. Figure 1b shows the ζ potential (measured with a Zetasizer Nano ZS, Malvern, Worcestershire, UK) for our TiO${}_{2}$NPs as a function of pH, while the red horizontal line marks the stability limit [51] and shows that for pH ≥ 4, the suspension is stable (corresponding to ζ-potential values ⪅ 30 mV). This result therefore confirms the stability of the sample at neutral pH.

#### 2.4.2. Dynamic Light Scattering

A quantitative measure of the clustering in the suspension is obtained by measuring the hydrodynamic radius of the NPs, $R_{h}$, by dynamic light scattering (DLS) [52,53,54] (DynaPro Protein instrument, Wyatt Technology, Santa Barbara, CA, USA). This measurement reflects not only the size of the particle core but also any surface structure, as well as the type and concentration of any ions present in the medium. Figure 1c shows the normalized autocorrelation $g^{\left(2\right)}$ curves of a sample consisting of single size particles (monodisperse sample) obtained from the intensity fluctuations of the laser integrated in the instrument (λ = 680 nm) due to TiO${}_{2}$NP scattering for pH = 2 … 7 (note that TiO${}_{2}$ does not absorb at this wavelength). These fluctuations are random and related to the diffusion coefficient $D_{s}$, i.e., the $R_{h}$ of the particles undergoing Brownian motion [55]. A shift in the slope of the response towards longer time delays $\tau_{DLS}$ reflects a slower motion of the TiO${}_{2}$-NPs in solution, i.e., a larger $D_{s}$.

Small nanoparticles ($R_{h}=60$ nm) show a shorter delay ($\tau_{DLS}\approx 1$ ms) than large ones ($R_{h}$ > 1 μm), corresponding to $\tau_{DLS}\approx 10$ ms. The slower motion (Figure 1c), observed for 1≤ pH ≤3, is associated with the largest mean $R_{h}$ values ($R_{h}>350$ nm) with high dispersion (Figure 1d), indicating the presence of clusters. The diffusion coefficient is the same for all pH ≥4, giving $R_{h}\approx 60$ nm with low dispersion. These results are compatible with the obtaining of a stable suspension (−50 mV <ζ<−30 mV for pH ≥4), as opposed to the large fluctuations in size—associated with large values of $R_{h}$ (clustered sample)—for 1≤ pH ≤3. Thus, the information provided by the DLS-based measurements corroborates that provided by the ζ potential.

## 3. Experimental Setup

### 3.1. Optical Setup

Figure 2a shows the experimental setup. A Q-switched, frequency-tripled Nd:YAG laser, pulsed at a repetition rate of $\nu_{p}$ 10 Hz is used to pump an Optical Parametric Oscillator (OPO) tuned to the absorption of FITC (λ = 490 nm). Additional details are available in the Appendix A.

Pulse-to-pulse stability of the signal emitted by the OPO requires uninterrupted operation, despite the built-in ability to program an arbitrarily short pulse train. Under optimal conditions, the OPO output will be pulses with energy $E_{p}=\left(7.5\pm 0.5\right)$ mJ, with a duration of $\tau_{p}\approx 5$ ns and a repetition rate of $\nu_{p}=10$ Hz. Each individual pulse is monitored and recorded during the experiment by a Si photodiode (DET10A2, Thorlabs, P1 in Figure 2a, $\tau_{P1}=1$ ns rise time and responsivity $S_{P1}=0.2$ A/W @ λ=500 nm), which receives a small part of the pulse through a beam pick-off. The OPO beam is astigmatic with horizontal (<0.7 mrad) and vertical (3 to 9 mrad) divergences specified by the manufacturer. In order to minimize the energy loss due to changes in the optics and to maintain beam quality, we minimized the number of optical elements in the beam path in front of the experimental cell.

Beamsplitter S (ratio 15:85) allows individual pulse monitoring on P1 (Figure 2a). The pulse energy is adjusted, for detector protection and optimality by two Neutral Density (ND) filters: an absorbing Kodak Wratten II (Rochester, NY, USA), with optical density OD=2.0 and a reflective N-BK7 filter (ND30A, Thorlabs, Newton, NJ, USA) with OD=3.0. The detector signal is fed to a 2.5 GHz digital oscilloscope (WaveRunner 625Zi, LeCroy, New York, NY, USA) coupled at 50 Ω. By calibrating P1, we record the energy of each pulse sent to the FITC sample.

The beam transmitted by S, with pulse energy in the range 100 μJ–3 mJ, controlled by an adjustable set of ND filters F1 (Kodak Wratten II), is reflected by a low-pass ($\lambda_{cutoff}=500$ nm, $45^{\circ }$ dicroic mirror DM (FF500-Di01-25x36, Semrock, Buffalo, NY, USA) and focused onto the sample by a f=75 mm lens, L1. The astigmaticity of the laser beam and the variability from one pulse to the next make it difficult to estimate the surface energy density on the sample. Estimating the focused beam to be rectangular in size 200×500μm${}^{2}$, the resulting energy density is in the range of 1…30 mJ/mm${}^{2}$. Because of the uncertainty in these estimates, all experimental results are given in units of pulse energy.

The backscattered fluorescence pulses emitted in response to each excitation are collected by L1 (estimated NA = 0.17), spectrally filtered by the dichroic element, DM, to remove residual energy at the pump wavelength, attenuated (if necessary) by the set of ND filters F2 (Kodak Wratten II) and focused on the detector by a second lens L2. The fluorescence is detected either by a fast photomultiplier detector (H10721-210, Hamamatsu, Shizuoka, Japan, rise time $\tau_{rt}=0.57$ ns and sensitivity 0.1 A/W @ λ=500 nm) or spectrally analyzed by a spectrometer (USB 2000, OceanOptics, Orlando, FL, USA, optical FWHM resolution 1.5 nm). Both detection systems are fiber coupled (QP-200-2-UV-BX, OceanOptics, core diameter 200 μm, SMA905 adapter, NA = 0.22) through a matched focusing f= 8 mm lens (NA = 0.55) L2.

### 3.2. Synchronization

The acquisition system is designed with the following features:1.The ability to monitor the variability of each OPO pulse and correlate it with the observed fluorescence emission;2.Controlled number of pulses with sequential acquisition;3.Constant duration for each pulse sequence;4.Compatibility with the memory depth of the oscilloscope (segmentation).

The first point is necessary for post-treatment in cases where pulse-to-pulse variability may mix nominally different pulse energies, thus allowing meaningful comparison of the observed fluorescence. The second point arises from the presence of photobleaching (Section 3.4) and the need to maintain compatible experimental conditions. While the information is contained in a short time interval (about 20 ns) around each pulse, there is a long delay (0.1 s) during which the detectors collect only noise. If the oscilloscope were to acquire continuously, its memory would be saturated with a single pulse. The need for immediate storage would result in a long treatment time, which would prevent the acquisition of a sequence of pulses at the laser cadence. As a result, the experimental conditions may not be the same from pulse to pulse (and from sample to sample) due to the possible difference in treatment and storage time. Using the oscilloscope’s segmentation mode, it is possible to store the essential parts of the time traces, thus realizing comparable experimental sequences, saving acquisition time, concentrating on the important information and avoiding wasting laser energy (the laser is blocked by the shutter when not needed to pump the sample because switching it off would introduce thermal transients in the OPO and poorer pulse-to-pulse reproducibility).

For quantitative measurements, careful control of the number of pulses and their synchronized detection are required. The mechanical shutter at the output of the laser not only prevents photobleaching of the FITC, but also ensures the synchronization of all operations (Figure 2b), particularly as regards the spectrometer. Control is achieved by home-made electronics and is based on the synchronization signal from the Q-switched laser (a TTL signal of 5 ms duration, red chronogram). The spectrometer cannot synchronize directly to the TTL signal due to an internal delay (1 μs) that precedes the beginning of the acquisition, while the laser pulse is delayed by 150 ns relative to the TTL signal (Laser Pulse inset in Figure 2b). To ensure the proper acquisition of the optical spectrum, the spectrometer is opened 1 μs after the shutter opens, which in turn precedes the TTL signal (5 ms duration) by 20 ms. The spectrometer acquisition window (green chronogram) lasts 80 ms.

### 3.3. Sample Mounting

The prepared homogeneous solutions of FITC and TiO${}_{2}$-NPs are placed in a cell with a diameter of $d_{c}=10$ mm, formed by a microscope slide on one side and a #1 coverslip on the other. The thickness of the cell is $t_{c}=2$ mm, controlled by the superposition of four 500 μm spacers (cat. #70366-13, Electron Microscopy Sciences, Hatfield, PA, USA). This choice results from the need for a sufficiently thick sample on the one hand, and from the possibility of reducing dye photobleaching [1] in the pumped volume thanks to convective motion in the fluid [56] (the cell is also mounted in a vertical configuration so that gravity enhances convection).

### 3.4. Determination of the Optimal Acquisition Time-Window

As FITC is subject to photobleaching [57], we need to characterize the fluorescence decay in response to prolonged exposure. This in turn determines the number of pulses to which we can expose the sample before bleaching occurs. Sequences of fluorescence spectra were collected for 1200 pulses (2 min) for each pair of experimental parameters ($C_{N}$, $E_{p}$). We observe a sharp fluorescence decrease for all NP concentrations (Figure 3), which complicates the measurement. In order to enable a statistically significant sample—due to fluctuations in the pump pulse energy—while containing the influence of photobleaching, we use a 1 s time window (10 pump pulses) as a reasonable compromise for all pairs of experimental parameters ($C_{N}$, $E_{p}$). Measurements are therefore made on a fresh, unused sample that is exposed to only 10 pulses before being replaced.

### 3.5. Energy Measurements

In order to run the OPO in the optimal, stablest mode of operation, we keep the energy of the UV pulses, issued from the Spectra Physics pulsed laser, constant and attenuate with the help of calibrated filters (cf. Appendix A) the energy impinging on the fluorescent sample. To account for all losses along the optical path, the energy meter (PE25BF-C, Ophir) is positioned in place of the cell, and the energy $<E{>}_{Energy-Meter}$ recorded (averaged over 100 pulses at $\lambda_{P}=490$ nm) is compared with the energy $<E{>}_{P1}$ measured by P1 (also averaged over 100 pulses):(1)$$<E{>}_{P1}=\frac{1}{100}\sum_{i=1}^{100}\frac{A_{i}^{90\%}}{R_{P1}R_{osc.}}\;,$$
where $R_{P1}$ is the responsivity of P1 (0.2 A/W @ $\lambda_{P}=490$ nm), $R_{osc.}$ is the input impedance of the oscilloscope (50Ω), and $A_{i}^{90\%}$ is the area of a pulse at 90% of its maximum height (measured in V · s), calculated from the individual time traces.

The results are shown in Figure 4, which shows in its upper panel the experimental mean values with standard deviation on both axes, resulting from fluctuations in the energy of the pump pulses. The lower graph shows the deviation (in percent) between the actual measurement and the best linear fit obtained in the upper panel.

Thus, the calibration plot (Figure 4) also shows the fluctuations of the pulse energy arriving at the experimental sample as a function of the measured (fluctuating) signal at P1 according to
(2)$$<E{>}_{Energy-Meter}=10^{-OD}\cdot A\cdot <E{>}_{P1}$$

## 4. Data Analysis

The analysis of the amplification signals was carried out according to the following procedure:1.For all combinations of ($C_{F},C_{N}$), we measure all quantities of interest for six different (nominal) values of pulse energy $E_{p}=\left(0.15,0.30,0.60,1.20,2.00,3.00\right)$ mJ (the actual values shown in the graphs are adjusted based on the reference measured by P1).2.For each energy and preparation ($C_{F},C_{N}$), we repeat the measurements on six independent samples.3.For each energy and sample, we acquire and record 10 consecutive fluorescence spectra obtained from 10 pump pulses (1s total acquisition time).4.For each energy and sample, we compute the mean $\bar{X}$ and the standard deviation $\sigma_{X}$ of the measured quantities $X_{i}$ (fluorescence amplification, gain, fluorescence decay time and Full Width at Half-Maximum (FWHM) of the measured spectra) for 10 measurements.5.For each measured quantity, we calculate the weighted average $\bar{Y}$ and the standard deviation $\sigma_{Y}$ over the six repetitions (samples) [58]:
(3)$$\bar{Y}=\frac{1}{M}\frac{\sum_{n=1}^{M}\bar{X_{n}}/\sigma_{X_{n}}^{2}}{\sum_{n=1}^{M}1/\sigma_{X_{n}}^{2}}\;\;\;\sigma_{Y}^{2}=\frac{1}{M-1}\frac{1}{\sum_{n=1}^{M}1/\sigma_{X_{n}}^{2}}\;,$$
where $\bar{X_{n}}$ represents any of the measured averages, $\sigma_{X_{n}}$ its standard deviation and *M* is the number of repetitions (M=6 throughout the experiment).

## 5. Results and Discussion

### 5.1. Influence of TiO${}_{2}$-NPs upon FITC Fluorescence Intensity

The addition of increasing concentrations of TiO${}_{2}$-NPs ($C_{N}$ from 1.56 to 6.25 mg/mL) to a 200 μM solution of FITC produces a monotonous growth in the collected fluorescence intensity spectra (Figure 5a) at the nominal pulse energy $E_{P}=3$ mJ. Plotting the spectral intensity maximum for all $C_{N}$ values as a function of pump energy $E_{P}$ (Figure 5b) shows a clear NP-induced amplification. A red shift in the position of the fluorescence maximum (from $\lambda_{M}=517$ nm at $C_{N}=0$ mg/mL to $\lambda_{M}=522$ nm at $C_{N}=6.25$ mg/mL) is evident and can be related to the longer optical path induced by the increased scatterer density. The energy dependence of the fluorescence intensity in the absence of TiO${}_{2}$-NPs is visible on a larger scale (inset of Figure 5b). The superlinear fluorescence growth is consistent with amplification by stimulated emission.

### 5.2. Influence of TiO${}_{2}$-NPs upon FITC Fluorescence Spectra

Accompanying the enhancement, a spectral narrowing of the collected fluorescence is observed as illustrated by the shape of the normalized spectra (Figure 6a) for the different TiO${}_{2}$-NPs, measured at the nominal energy $E_{P}=3$ mJ. Figure 6b shows the evolution of the FWHM as a function of the pump energy $E_{P}$: a monotonic reduction as a function of $E_{P}$ is observed for each concentration $C_{N}$, as well as a progressive reduction with eventual saturation at FWHM ≈5 nm when varying $C_{N}$ at fixed $E_{P}$. In the absence of NPs (blue curve), the FWHM remains reasonably constant (FWHM ≈20 nm) for all pump energy values in the studied range.

### 5.3. Influence of TiO${}_{2}$-NPs upon FITC Fluorescence Pulse Duration

The pump pulse has a duration $\tau_{p}$ comparable to the fluorescence time decay $\tau_{f}\approx 4$ ns [40]. Thus, the statistical expectation of one photoemission per pump pulse for each fluorophore strongly limits the amount of emitted fluorescence. Stimulated emission instead occurs on time scales much shorter than $\tau_{f}$ (by up to six orders of magnitude) and prepares the emitter for a new cycle well within the pulse duration $\tau_{p}$, allowing the emission of many photons per pump cycle (per emitter) and a greater overall photon yield.

In the non-stimulated emission regime, the fluorescence pulse duration $\tau_{fp}$ results from the convolution of the pulsed excitation and the molecular relaxation probability, leading to $\tau_{fp}>\tau_{p},\tau_{f}$. The almost instantaneous stimulated relaxation removes the influence of $\tau_{f}$, indicating a decrease in the value of $\tau_{fp}$. However, the limit $\tau_{fp}=\tau_{p}$ cannot be reached because the photon flux in the pulse wing falls below the rate required to sustain the stimulated emission, giving way to the standard fluorescence process (with its consequent extension of $\tau_{fp}$).

These basic considerations are confirmed by the measurements of Figure 7a, which shows the fluorescence pulse duration $\tau_{fp}$ measured by the fast photomultiplier as a function of $E_{P}$ for the three NP concentrations. The pulse width is obtained from the zero crossing of the normalized autocorrelation function of the signal, which contains a small contribution from the not entirely negligible detector response time (see Section 3). It is also important to note that the fluorescence pulse from which $\tau_{fp}$ is extracted is the convolution of all emission processes integrated over different *sources* (i.e., fluorescent molecules in the sample emitting independently) and is collected in the solid angle corresponding to the numerical aperture of the optics (NA= 0.17).

The characteristic fluorescence pulse width $\tau_{fp}$ decreases monotonically as a function of $C_{N}$ for all pump pulse energy values $E_{N}$, starting from ≈8.5 ns and reaching ≈7.0 ns at $C_{N}=6.25$ mg/mL (cf. inset of Figure 7a measured for 3 mJ pulse energy). At low pump energy, instead, there is hardly any change in $\tau_{fp}$, consistent with the previous discussion. Note that the decrease in $\tau_{fp}$ appears to be a *threshold* phenomenon, since *in the presence of NPs* it first undergoes a sharp decrease (when $E_{P}=300\;\mu$J →$E_{P}=1.2$ mJ), while its subsequent evolution ($E_{P}>1.2$ mJ) is gradual. This abrupt change supports an interpretation of the observations based on the onset of stimulated amplification and is reinforced by the increase in fluctuations in $\tau_{fp}$ accompanying the abrupt change (for the concentrations of 1.56 and 3.12 mg/mL), as is typical of phase transitions (here from a spontaneous to a stimulated process).

### 5.4. Influence of TiO${}_{2}$-NPs upon Optical Gain

The previous results have shown the main physical features of the amplification process introduced and controlled by the presence of scatterers. However, from a practical point of view, it is interesting to know how much advantage can be gained from the presence of NPs compared to the fluorescence yield of the fluorophore alone. For this purpose, we introduce a gain quantity defined as the ratio between the collected spectral peak fluorescence intensity in the presence of NPs and the same quantity in the absence of NPs, under the same illumination conditions:(4)$$G=\frac{I_{F,C_{N}}\left(E_{P}\right)}{I_{F,0}\left(E_{P}\right)}\;.$$

The gain is shown in Figure 7b for the three concentration values $C_{N}$. The blue data are those taken in the absence of NPs—hence, $I_{F,0}\left(E_{P}\right)$—which are used as the reference value and therefore correspond to G=1.

It is interesting to note that the overall shape of the gain curves (Figure 7b) is different from the fluorescence intensity curves of Figure 5b. The latter show superlinear behavior, with a *slow start* as a function of $E_{P}$, whereas G shows a *fast growth* that slows down with increasing pump pulse energy (and perhaps indicates the presence of saturation, at least for the black line). The contrasting behavior reflects the conceptually different nature of the quantities being plotted: while the fluorescence intensity $I_{F}$ shows the absolute amount of light collected, gain G quantifies the benefit derived from the addition of NPs to the fluorescent sample. The experimental results show that the gain is much stronger at lower pump energies than at higher ones, while it grows monotonically with $C_{N}$.

This is good news for many applications (including potential biological ones) since half of the total gain can be obtained at less than 1/3 of the maximum pump pulse energy, in the range we have explored. Two different strategies are therefore possible, depending on the scope of the application. In order to maximize the amount of collected fluorescence light, it will be worthwhile to increase the pump pulse energy, whereas to obtain the greatest benefits in terms of gain efficiency (without reaching the maximum value of G), it will be sufficient to use $E_{P}<1$ mJ, thus saving energy and limiting possible photodissociation, photobleaching and phototoxic effects.

Finally, it is important to note that the absolute amount of gain obtained with this setup is considerable. Figure 7b shows that it is possible to achieve G≈40 under the best experimental conditions we used ($E_{P}\approx 3$ mJ and $C_{N}=6.25$ mg/m*ℓ*). However, if lower scattering densities are preferred, it is still possible to obtain G≈10 ($C_{N}=1.56$ mg/m*ℓ*), which represents a substantial gain from an experimental point of view since an increase of one order of magnitude clearly raises a weak signal well above the background noise.

### 5.5. Additional Evidence

Additional evidence for the role played by the scatterers is provided by the photobleaching curves. Figure 3 shows the strong impact of the presence of TiO${}_{2}$-NPs on the fluorescence efficiency. Equivalent measurements conducted at lower FITC density (Figure 8) give even stronger evidence. By progressively increasing the NP concentration, from 0 to its maximum value, photobleaching progresses more and more rapidly as a function of time. This clearly indicates the role played by the TiO${}_{2}$-NPs in the degradation of the fluorescent molecule: the higher the NP concentration, the larger the number of fluorescent cycles per pulse. Thus, we can conclude that the role of the NPs is to increase the number of cycles and, with it, the fluorescence yield per laser pulse.

## 6. Conclusions

The results of this investigation show that by adding TiO${}_{2}$-NPs to the sample, it is possible to obtain fluorescence amplification by stimulated emission with FITC (a common and FDA approved fluorescent dye [26]) with a gain of up to a factor of 40. For comparison, amplification is obtained at concentrations five times lower than those of Rh6G commonly used for random lasing in dead tissue and in an aqueous medium at a neutral pH [27].

The stimulated amplification is evidenced by the observation of an increased fluorescence yield, a significant reduction in linewidth (up to four or five times) and a shorter fluorescence pulse duration. In applications, the optical linewidth reduction can be used to provide improved detection when different fluorophores are used simultaneously to label different agents. Optical amplification is achieved at sufficiently low pulse energies, limiting photodissociation, photobleaching or phototoxic effects. Other types of NPs can be considered (e.g., ZnO, silver or gold), depending on the experimental conditions (e.g., absorption wavelength, scattering efficiency, NP size, and sample requirements).

The strong amplification of a fluorescent signal paves the way for numerous applications in biology, environmental sensing, chemical detection, food safety, to name but a few. The greatly increased signal strength allows the detection thresholds to be lowered, enabling contamination, pollutants or generally low populations to be easily identified using the same detection chain. Wavelength multiplexing is facilitated by the significant narrowing of the fluorescence line, as the spectral signatures of the different fluorophores become more easily identifiable. In addition, crosstalk between the short-wavelength emission tail and the long-wavelength reabsorption tail—i.e., self-quenching—is greatly reduced. Finally, the nanosecond timescales of pulsed fluorescence can be exploited for time-resolved monitoring.

## Figures and Tables

**Figure 1 nanomaterials-13-02875-f001:**
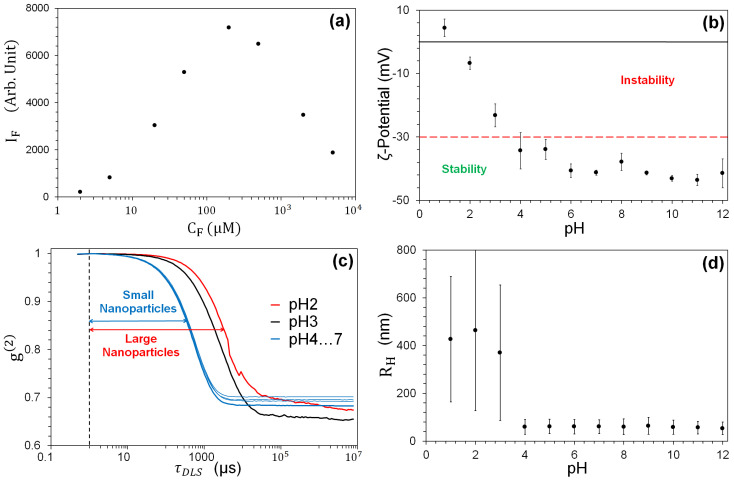
(**a**) Fluorescence intensity *I*${}_{F}$ as a function of $C_{F}$. *I*${}_{F}$ is obtained from the maximum ($\lambda_{F}\approx 520$ nm) of each spectrum measured by the spectrofluorimeter. (**c**) Normalized scattering intensity autocorrelation $g^{\left(2\right)}$ for different pH values. (**b**,**d**) The TiO${}_{2}$-NP stability in H${}_{2}$O mQ as a function of pH for 6 independent samples. (**b**) ζ-potential measurements. (**d**) Mean hydrodynamic radius $R_{h}$ (dots). The bars denote the size dispersion.

**Figure 2 nanomaterials-13-02875-f002:**
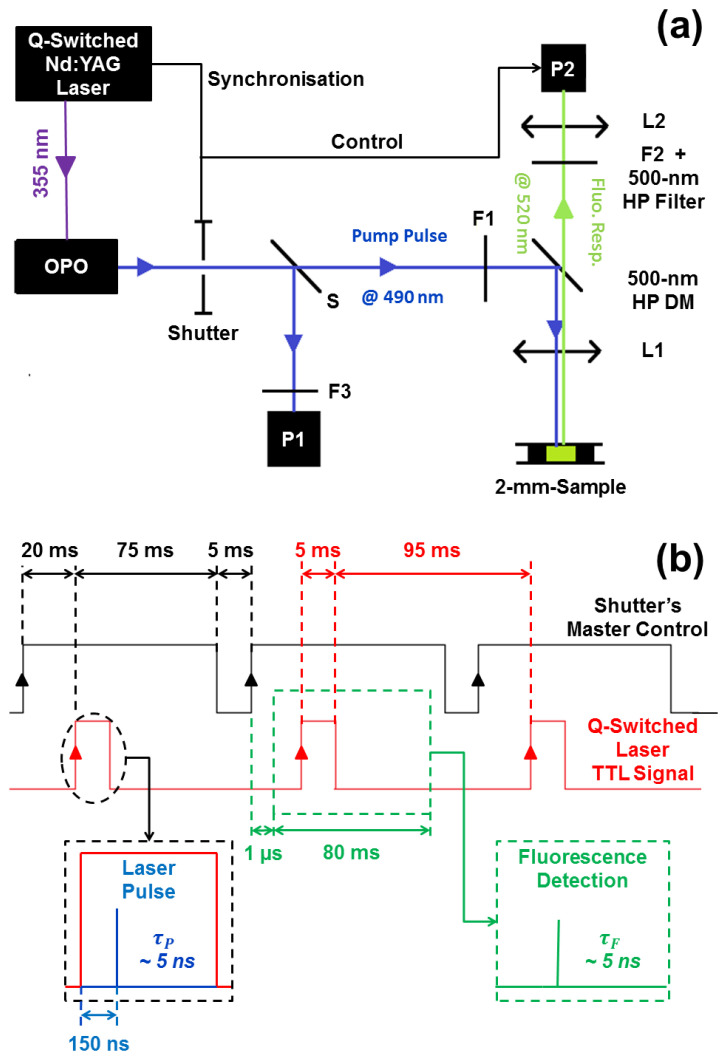
(**a**) Experimental setup. The pump laser consists of a frequency-tripled ($\lambda_{t}=355$ nm) Nd:YAG pumping a tunable OPO. The resulting $\lambda_{P}=490$ nm pulses are split by a beamsplitter S, which captures a small fraction of the energy for monitoring on photodiode P1 (F3 is a set of protective ND filters for P1). The transmitted beam is attenuated by a variable set of ND filters F1 and then deflected by a high-pass ($\lambda_{cutoff}=500$ nm) dichroic mirror DM, and finally focused by lens L1 (f=75 mm) onto the sample placed at the focus of L1. The backscattered fluorescence pulses are collected by L1 (estimated NA = 0.17) and detected by either a spectrometer or a time-resolved detector (both denoted P2). An ND filter F2 can be inserted to attenuate high fluorescence signals to avoid the saturation of P2. The mechanical Shutter determines the length of the pulse train sent to the experiment. (**b**) Shutter-controlled (black chronogram) acquisition of fluorescence spectra (and fluorescence pulses). Synchronization signal from the laser (red chronogram). Spectrometer control (green) and laser pulse delay control (blue).

**Figure 3 nanomaterials-13-02875-f003:**
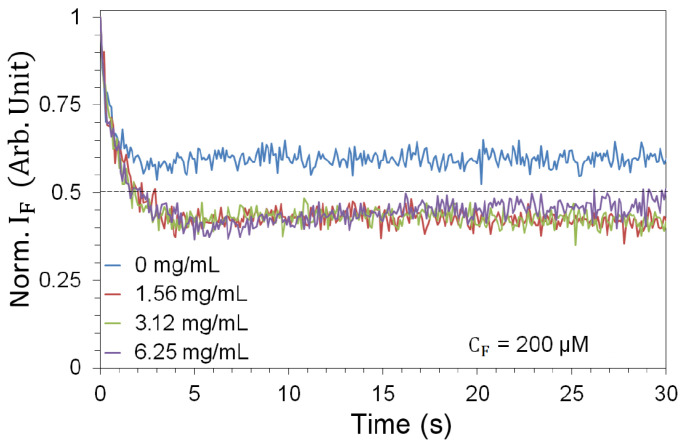
Photobleaching, shown over a 30 s measurement window (300 laser pulses), in the absence (blue) and in the presence of TiO${}_{2}$-NPs at 3 mJ pump energy (most extreme photobleaching). The curves are averages taken over 6 different samples.

**Figure 4 nanomaterials-13-02875-f004:**
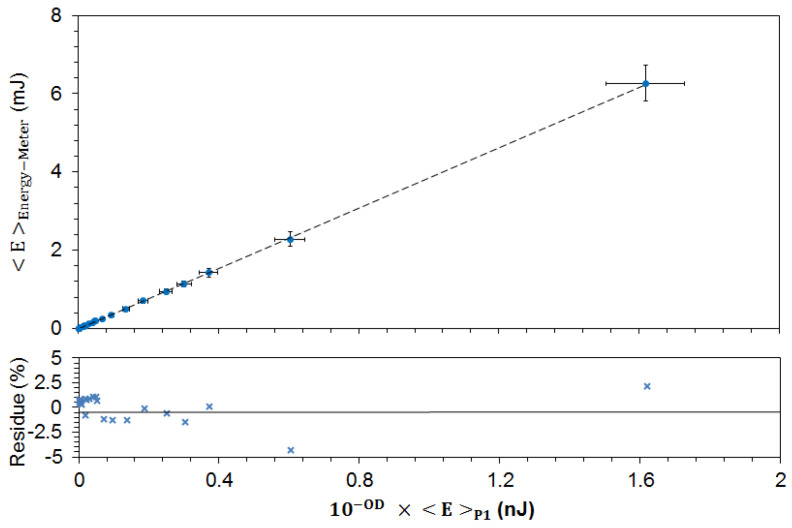
Calibration plot of the energy arriving at the sample as a function of the energy measured by the monitoring photodiode P1, for the different combinations of the ND filters F1. All measurements are averaged over 100 pump pulses.

**Figure 5 nanomaterials-13-02875-f005:**
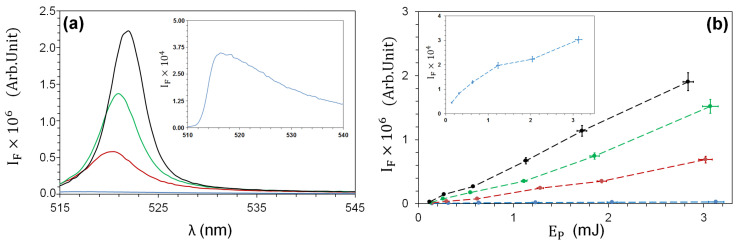
(**a**) Fluorescence emission spectra at pump energy $E_{p}=3$ mJ; (**b**) maximum of fluorescence intensity as a function of the pump energy $E_{P}$ for $C_{N}=$: 0 (blue); 1.56 (red); 3.12 (green); 6.25 (black) mg/mL. Insets: fluorescence spectrum (**a**) or intensity (**b**) at $C_{N}=0$ mg/mL on an expanded vertical scale.

**Figure 6 nanomaterials-13-02875-f006:**
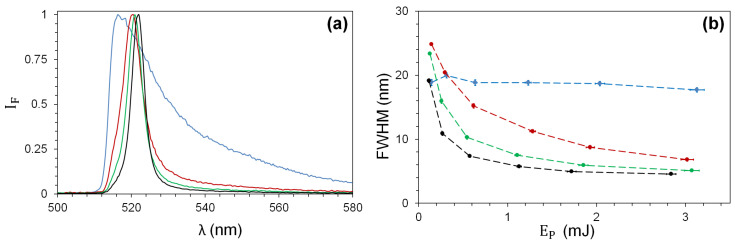
(**a**) Normalized fluorescence spectra at $E_{P}=3$ mJ and (**b**) FWHM of spectra as a function of the pump energy for $C_{N}$: 0 (blue); 1.56 (red); 3.12 (green); 6.25 (black) mg/mL.

**Figure 7 nanomaterials-13-02875-f007:**
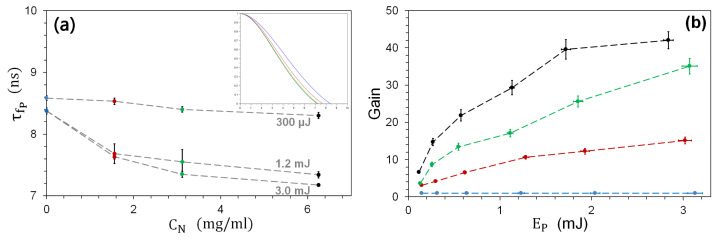
(**a**) Fluorescence pulse duration $\tau_{fp}$ as a function of $C_{N}$ for 3 different pump energies $E_{P}=$: 300, 1200 and 3000 μJ. The inset shows the pulse’s autocorrelation function, from which the pulse duration can be deduced [59] (crossing of the horizontal axis). (**b**) Fluorescence gain for $C_{N}=$: 0 (reference measurement, blue); 1.56 (red); 3.12 (green); 6.25 (black) mg/mL.

**Figure 8 nanomaterials-13-02875-f008:**
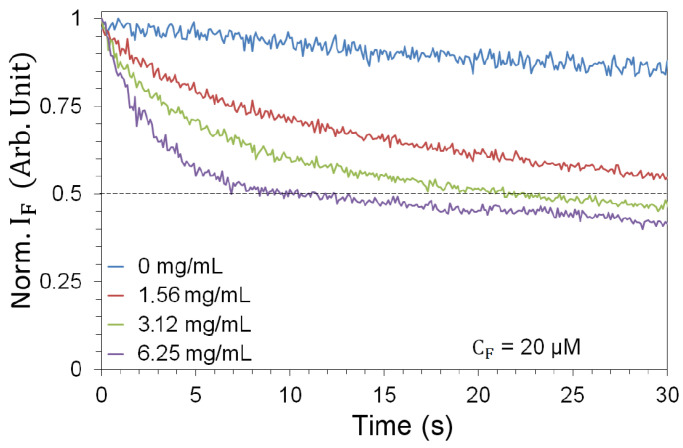
Photobleaching, shown over a 30 s measurement window (300 laser pulses), in the absence (blue) and in the presence of TiO${}_{2}$-NPs at 3 mJ pump energy for $C_{F}=20\;\mu$M. The curves are averages taken over 6 different samples.

## Data Availability

The data presented in this study are available upon reasonable request from the corresponding author.

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
