# Peer review of "Nanoscatterer-Assisted Fluorescence Amplification Technique"

_nanomaterials, 2023, doi:10.3390/nano13212875_

Round 1
Reviewer 1 Report
Comments and Suggestions for Authors
Manuscript Number: nanomaterials-2647538
Title: Nanoscatterer-Assisted Fluorescence Amplification Technique
Authors: Sylvain Bonnefond, et al.
Recommendation: Minor revisions
Reviewer comments: (Express Comments to the Author):
The authors reported a broadband, geometry-independent and flexible feedback scheme based on the random scattering of dielectric nanoparticles allows the amplification of a fluorescence signal by partial trapping of the radiation within the sample volume. Therefore, I recommend it for publication after minor revisions.
Comments:
1 In Part Materials and Methods, "the refractive index of the NanoParticles (NPs) must be compared with that of this medium.", how is the above content reflected in the experiment?
2 In Part 2.4.2. Dynamic light scattering, why was the "680nm laser" chosen?
3 In Part Conclusion, "In particular, the optical linewidth reduction can provide improved detection when different fluorophores are used simultaneously to label different agents.", how is the above reflected in the passage?
Comments on the Quality of English Language
That is ok.
Reviewer 2 Report
Comments and Suggestions for Authors
In this paper, the authors considered the technics for amplification of fluorescence based on the random scattering of dielectric nanoparticles. In my opinion, this paper have chance to be published in present form. However, I recommend to improve introduction part. Recently a lot of papers, published about fluorescence amplification due to internal resonance of nanoparticles. Please present the review. For example, you can find this paper: doi: 10.1039/b802918k
Reviewer 3 Report
Comments and Suggestions for Authors
The authors report on titania nanoparticles to enhance the fluorescence of FITC dye due to the amplification by stimulated emission or random lasing. While the random lasing with TiO2 NPs and other dyes are well known for many years, the manuscript presents new results for the case of FITC, which is important for bioimaging. The manuscript is well written, and the results are interesting. But it contains some critical points, which affect the scientific soundness of the presented results. The authors are invited to answer the following questions and comments related to the scientific and technical aspects of the manuscript.
1. What is the main scientific achievement of the paper in comparison with Refs.21-26 ?
2. It was underlined “Amplification of up to a factor of 40” in the abstract. However, the amplification for the random lasing depends on many factors as system geometry, absorption losses and pump energy. Maybe it would be better to mention the amplification conditions in more detail in the abstract?
3. Why terms as “random laser” or “random lasing” are not used in the abstract and key words?
4. Wording “the Ambartsumyan team” (line 36) is better to change on “the Ambartsumyan and co-workers” because both N.Basov (the Nobel Prize winner) and V.Letokhov were co-authors of Ref.24 .
5. Because the strong scattering of exciting laser radiation can also contribute to the enhanced absorption of fluorescent molecules its contribution should be discussed together with the random lasing. The excitation light scattering in a semiconductor nanostructure was found to provide an enhancement of the luminescence without random lasing (see for example: Nanoscale Research Letters 2012, 7:5244).
6. Since the excitation pulses were strong and the irradiated volume was limited the possible role of laser induced heating should be discussed.
7. What is the main advantage of the used registration scheme? Is it possible to prove the benefit of the used time-gated technique in comparison with conventional spectroscopic registration? Otherwise, the methodological part of the manuscript including Figs.1,3,4 can be moved to the section of Suppl. Materials.
8. Why the fluorescence spectra of FTIC without TiO2 NPs (figures 5(a) and 6(a)) are strongly asymmetric and don to centered at 520-530 nm in contrast to the reported in the literature? (see for example: Scientific Reports | 7: 17484 | DOI:10.1038/s41598-017-17459-y)\
9. The reference list can be improved by skipping reference doubling (Refs. 26 and 33) and adding new ones related to the possible role of the excitation light scattering and FTIC spectroscopy for biophotonic applications.
Round 2
Reviewer 3 Report
Comments and Suggestions for Authors
The authors' responses are reasonable and satisfactory. The revised manuscript may be accepted for publication.